# Solvent Effect on the Structure and Properties of RGD Peptide (1FUV) at Body Temperature (310 K) Using Ab Initio Molecular Dynamics

**DOI:** 10.3390/polym13193434

**Published:** 2021-10-07

**Authors:** Khagendra Baral, Puja Adhikari, Bahaa Jawad, Rudolf Podgornik, Wai-Yim Ching

**Affiliations:** 1Department of Physics and Astronomy, University of Missouri-Kansas City, Kansas City, MO 64110, USA; kbx67@mail.umkc.edu (K.B.); paz67@umkc.edu (P.A.); bajrmd@mail.umkc.edu (B.J.); 2School of Physical Sciences, Kavli Institute of Theoretical Science, University of Chinese Academy of Sciences, Beijing 100049, China; rudipod@gmail.com; 3CAS Key Laboratory of Soft Matter Physics, Institute of Physics, Chinese Academy of Sciences, Beijing 100090, China; 4Wenzhou Institute of the University of Chinese Academy of Sciences, Wenzhou 325000, China; 5Department of Physics, Faculty of Mathematics and Physics, University of Ljubljana, SI-1000 Ljubljana, Slovenia

**Keywords:** RGD peptide (1FUV), ab initio molecular dynamics, total bond order, partial charge, dielectric function

## Abstract

The structure and properties of the arginine-glycine-aspartate (RGD) sequence of the 1FUV peptide at 0 K and body temperature (310 K) are systematically investigated in a dry and aqueous environment using more accurate ab initio molecular dynamics and density functional theory calculations. The fundamental properties, such as electronic structure, interatomic bonding, partial charge distribution, and dielectric response function at 0 and 310 K are analyzed, comparing them in dry and solvated models. These accurate microscopic parameters determined from highly reliable quantum mechanical calculations are useful to define the range and strength of complex molecular interactions occurring between the RGD peptide and the integrin receptor. The in-depth bonding picture analyzed using a novel quantum mechanical metric, the total bond order (TBO), quantifies the role played by hydrogen bonds in the internal cohesion of the simulated structures. The TBO at 310 K decreases in the dry model but increases in the solvated model. These differences are small but extremely important in the context of conditions prevalent in the human body and relevant for health issues. Our results provide a new level of understanding of the structure and properties of the 1FUV peptide and help in advancing the study of RGD containing other peptides.

## 1. Introduction

Biomolecular materials containing the arginine-glycine-aspartate (RGD) sequence are always at the center of biophysics research in their application such as in the bone scaffold, synthesis, and regeneration of tissue and cartilage [1,2], in imaging as radiotracers [3,4,5,6], for cancer therapy [7,8,9], and in targeted drug delivery [10,11,12,13,14,15]. The RGD motif peptide serves as a primary integrin recognition site in extracellular matrix proteins since it has a strong binding affinity to integrins, which are heterodimeric cell surface receptors and mediate cell-extracellular matrix adhesion [16,17,18,19]. Out of 24 known integrins, one third of them bind to the RGD motif as the primary recognition sequence in their ligands, which makes RGD an attractive target in numerous drug delivery systems [20,21,22,23,24]. The increased application of integrins in drug development and their functions in the physiological processes requires a complete understanding of the structure and properties of the RGD peptides at body temperature (310K), crucial in the design of selective inhibitors. Therefore, a detailed study of the structure and properties of 1FUV in RGD peptides at 310 K is of particular interest in delineating the modifications that occur with a rise or fall of body temperature. Moreover, the short- and long-range interactions are crucial for the molecular recognition and self-assembly of biological macromolecules, and consequently determining the microscopic parameters such as partial charge and frequency-dependent dielectric functions based on more accurate quantum mechanical calculation will facilitate the fundamental understanding of electrostatic, polar, and van der Waals-London dispersion interaction in biological (macro)molecules.

The peptide-water interfacial reactions are of paramount importance in biological systems as solvation is ubiquitous in biological interaction with water as an inevitable component in blood and multicomponent body fluids. Understanding the peptide-water interaction is also essential regarding the RGD-targeted drug delivery mechanisms. It is, therefore, crucial to investigate the consequences of peptide-water interaction and aqueous solvation and its effects on internal bonding, charge distribution, and dielectric response. The solubility of 1FUV in water is generally attributed to the hydrogen bonding between water molecules and oxygen from the peptide backbone. To explore the details of peptide solvation properties, molecular dynamics (MD), molecular mechanics, and Monte Carlo methods, all based on specific force field parameters, are widely used. Nevertheless, the information on the role played by the hydrogen bond (HB) within RGD and at the RGD-water interface is in general still missing, as are the relevant ab initio MD (AIMD) studies [12,25,26,27]. Classical MD studies cannot provide a quantitative description of essentially quantum HBs (O–H and N–H) since they depend on the details of the force field parameterization. Consequently, the need for AIMD simulation in the study of dry and aqueous solvated peptide/protein over classical MD has been repeatedly recognized [28,29,30,31] and a more accurate AIMD calculation with a sufficiently large number of water molecules is highly desirable for a complete understanding of 1FUV aqueous solvation.

Although experimental techniques such as nuclear magnetic resonance (NMR), surface-enhanced Raman spectroscopy, X-ray photoelectron spectroscopy, etc., are used to investigate the structure and properties of RGD containing peptides [16,32,33,34,35], the experimental studies specifically involving 1FUV are scant. Moreover, although experimental studies can probe the consequent changes of the peptide structure due to water contact, they cannot provide critical information and details of microscopic properties and interfacial reaction mechanisms at the atomistic scale. Computer-assisted atomistic simulation thus seems the best alternative to investigate the molecular behavior and provide new insights to complement the meager experimental data [36,37], in many cases guiding the experiments as well. Nevertheless, only a few computational studies have so far been reported on the 1FUV peptide [38].

To answer the issues raised above and provide new insights into the study of the 1FUV peptide, this study is designed to simulate four models of dry and solvated 1FUV at 0 K and body temperature, namely 1FUV0, 1FUV310, 1FUVS0, and 1FUVS310, respectively. Our study provides a deeper understanding of the temperature-dependent structure, fundamental properties, peptide-water interfacial reaction mechanism, and solvation effect. To the best of our knowledge, our effort is the first example of a study dedicated to the 1FUV peptide using AIMD that ensures the reliability of the simulated structures and calculated properties. The rest of the paper is structured as follows. In the next section, a brief description of the modeling technique followed by the simulation workflow is described. The main part is the result and discussion section, in which we discuss our findings and articulate the future prospects for applying AIMD to other complex biomolecular systems.

## 2. Materials and Methods

### 2.1. Modeling 1FUV Peptide

The RGD peptides that specifically bind to *α_v_β_3_* and *α_v_β_5_* integrins have medical significance in designing the inhibitors of tumor and retinal angiogenesis [39,40], in osteoporosis [41], and in targeted drugs for tumor vasculature [42]. Considering the importance of *α_v_β_3_* and *α_v_β_5_* integrins [16] and as 1FUV binds only to these integrins, we choose to focus on 1FUV in this work.

The initial structure of 1FUV was taken from the Research Collaboratory for Structural Bioinformatics protein data bank (PDB) generated from the NMR measurements [43]. The stored PDB file contains about 19 such models with the same composition, the number of atoms, molecular weight, and the chemical formula, but slightly different molecular volumes. In this study, we chose the first model out of 19, which has 135 atoms comprised of 11 amino acids. The initial atomic positions of 1FUV were enclosed in a sufficiently large supercell of size 37.12 Å × 34.41Å × 28.95 Å to avoid any unintended interaction due to periodic boundary conditions. This structure was then fully relaxed by applying a density functional theory (DFT)-based Vienna ab initio simulation package (VASP) [44,45] known for its efficiency in structure optimization and energy minimization to obtain 1FUV0, the structure at 0 K. We used a projector-augmented wave method [46,47] with Perdew-Burke-Ernzerhof potential for the exchange correlation functional within the generalized gradient approximation [48]. We employed a relatively high cutoff energy of 500 eV and set the electronic convergence criterion at 10^−5^ eV. The force convergence criterion for ionic relaxation was set at 10^−3^ eV/Å. A single K-point calculation was used as the simulated models are sufficiently large supercells.

The fully relaxed structure at 0 K was then heated slowly to 310 K, equivalent to human body temperature, using NVT ensemble within 4 ps. A Nose thermostat was used to control the temperature of the heat bath [49,50]. A time step of 0.5 fs was used for the ionic motion during the simulation to ensure accuracy for integration of the equation of motion, especially for the light atoms such as hydrogen. The peptide model at this elevated temperature was then equilibrated for 5 ps to obtain the equilibrium structure at 310 K, viz. 1FUV310. Out of 5 ps run, the fluctuations of temperature and pressure were minimized by the initial 4 ps run, and the averaged output structure is taken from the last 1 ps run, thus used as a production run. The solvated model at 0 K (1FUVS0) was constructed based on 1FUV0 by surrounding the peptide structure with 80 water molecules. The 1FUVS0 was then again fully relaxed by using VASP. Similar to the dry model, this solvated structure was then heated to 310 K to obtain the 1FUVS310 model using NVT ensemble within 4 ps. The same protocol as used in the dry model was adopted to obtain the equilibrated structure of the solvated model at 310 K.

To reveal the structural stability of simulated models, we present the root-mean-square deviation (RMSD) of atomic positions of the peptide structure during equilibration at 310 K in Figure 1a. The time-RMSD profile of global structure shows a smooth increase followed by a leveling off at ~ 1.90 Å in the dry model and ~ 1.15 Å in the solvated model, respectively, indicating that the equilibrium has been reached. The constant RMSD shown in the inset for the last 1 ps run, used as a production run, ensures the sufficient equilibration of simulated structures. Figure 1b shows the velocity autocorrelation function (VACF), including exponential fitting, during the equilibration of dry and solvated structures at 310 K which confirms the loss of initial velocity from the previous configuration and attest to sufficient equilibration of the systems. It also implies that the amplitude of VACF is smaller in the solvated model and dies out quicker than that in the dry model. In addition, the exponential fitting shows the solvated model reaches equilibrium faster than the dry model.

### 2.2. Calculation of Properties

The AIMD simulated and VASP relaxed structures discussed above are then used as input for the electronic structures and dielectric response calculations using the in-house developed package of the orthogonalized linear combination of atomic orbitals (OLCAO) methodology [51]. The combination of these two DFT methods, VASP and OLCAO, is found to be robust and highly effective in studying the electronic structure and bonding properties, especially for complex biomolecules [52,53,54,55]. In recent years, we have further demonstrated that the combination of these two methods is extremely effective in dealing with new and emerging problems for complex protein structures related to SARS-CoV-2 [56,57,58,59,60].

The OLCAO is an all-electron method using local density approximation of DFT in which the Gaussian-type orbitals are employed for the expansion of the atomic basis set. There are many advantages of using OLCAO for electronic structure calculation, such as the flexibility of the basis sets choice, lower computational cost, and ease of bonding and charge transfer analysis using the Mulliken scheme [51]. The implementation of localized atomic orbitals in the basis expansion enables us to quantify the charge transfer and interatomic bonding via effective charge (*Q**) on each atom and bond order (BO) value between pairs of atoms (*ρ_αβ_*) using the Mulliken scheme [61,62] as
(1)$$Q_{\alpha }^{*}=\sum_{i}\sum_{n,occ}\sum_{j.\beta }C_{i\alpha }^{*n}C_{j\beta }^{n}S_{i\alpha ,j\beta }$$
(2)$$\rho_{\alpha \beta }^{}=\sum_{n,occ}\sum_{i,j}C_{i\alpha }^{*n}C_{j\beta }^{n}S_{i\alpha ,j\beta }$$
where *S_iα,jβ_* is the overlap matrix between the basis Bloch sums of the orbital index (*i*, *j*) and atomic specification (*α, β*). N is the band index, *i*, *j* are the orbital quantum numbers, and *C_jβ_* is the eigenvector coefficient.

From the calculated value of *Q** the charge transfer between the ions due to atomic interaction can be quantified in terms of the partial charge (PC). It is defined as the deviation of *Q** from the charge of neutral atom *Q*^0^ and is given by Δ*Q* = *Q*^0^ − *Q**. The BO value calculated from the above equation gives the direct quantitative measure of bond strength between a pair of bonded atoms. The sum of all BO values in the system is the total bond order (TBO), which is a quantum mechanically calculated parameter. It is a single quantum mechanical metric helpful in assessing the internal cohesion and strength in a material [63]. The use of TBO to characterize the internal strength of a material and correlate it with the calculated physical properties is a novel and highly appealing concept.

OLCAO is equally suitable to calculate the imaginary part, ε_2_ (ℏω), of the frequency-dependent complex dielectric function within the random phase approximation of inter-band optical transition theory according to the following equation:(3)$$\begin{array}{l} \varepsilon_{2}(\hbar \omega )=\frac{e^{2}}{\pi m\omega^{2}}\int_{BZ}dk^{3}{\sum_{n.l}\left|\langle \psi_{n}(\vec{k},\vec{r})|-i\hbar \vec{\nabla }|\psi_{l}(\vec{k},\vec{r})\rangle \right|}^{2}\times f_{l}(\vec{k})\left[1-f_{n}(\vec{k})\right]\\ \delta \left[E_{n}(\vec{k})-E_{l}(\vec{k})-\hbar \omega \right] \end{array}$$

## 3. Results

### 3.1. Analysis of 1FUV Structures

The RGD peptide, 1FUV, studied in this work contains 11 amino acid sequence ACDCRGDCFCG, namely alanine (Ala, A), cysteine (Cys, C), asparagine (Asp, D), arginine (Arg, R), glycine (Gly, G), and phenylalanine (Phe, F). As this amino acid sequence contains four C, this peptide is also known as RGD-4C, in which the cysteine position is at 2, 4, 8, and 10th place of the sequence, named as Cys1, Cys2, Cys3, and Cys4, respectively. Each Cys amino acid contains the S atom and the arrangement of two Cys, and hence the S–S bond plays an important role in the configuration and dipole moment of this peptide [64]. Although the presence of four Cys groups in RGD-4C allows three possible combinations of S-S bonds, only two natural configurations of this peptide exist based on the S–S bonding arrangement [16]. The peptide with Cys1–Cys4 and Cys2–Cys3 disulfide bonding arrangement is known as RGD-A isomer, while the one with Cys1–Cys3, Cys2–Cys4 bonding arrangement is known as RGD-B isomer [16]. As RGD-A is a far stronger binder to integrin *α_v_β_3_* than RGD-B [16], it is reasonable to explore details of the structure and properties of RGD-A.

The relaxed structures for RGD-A peptide (1FUV) at 0 K and 310 K in the dry and aqueous environment are shown in Figure 2. 1FUV0 shows only a minor change in its structure due to VASP relaxation compared with the original NMR structure [16] taken from the PDB. However, there are noticeable changes in its conformation after AIMD simulation at 310 K. It shows a rotation of amino acid groups and changes in the bond length (BL) and BO value, which will be discussed in detail later. In the case of the solvated model, there again appear notable changes in the peptide structure. More water molecules interact with the peptide and there is a rotation of amino acid groups at 310 K as compared with 0 K. In both dry and solvated models, the S–S bond containing amino acids come closer at 310 K as compared with those at 0 K. However, the average S–S BL increases and hence its strength decreases at 310 K. It is noticed that the configurations of HBs, O–H, change at 310 K as compared with 0 K, resulting in the overall decrease in its strength in the dry model. However, in the solvated model, a large contribution of O–H bonding strength comes from peptide-water interaction due to which its value increases at 310 K. These differences are, although small, important on issues related to human health.

### 3.2. Electronic Structure and Interatomic Bonding

Electronic structures are the fundamental properties of a material that help to understand many other physical properties. The calculated total density of states (TDOS) for the simulated models is shown in Figure 3. The top of the valence band is set at 0 eV. The overall features of TDOS at 0 K and 310 K in dry and solvated models look similar as they contain the same amino acid groups; however, there is a decrease in the highest occupied molecular orbital (HOMO)-lowest unoccupied molecular orbital (LUMO) gap with the increase in temperature. The position of the energy difference between HOMO and LUMO plays the main role in various chemical reactions. The number of peaks decreases in solvated models as compared with the dry model, but the overall intensity increases. The HOMO-LUMO gap for 1FUV0 is ~ 3.40 eV, while its value in 1FUV310 decreases to ~2.10 eV due to a temperature rise, indicating that the molecule is chemically more reactive at a higher temperature. The HOMO-LUMO gap decreases largely in solvated models as compared with dry models. In 1FUVS0, this gap is ~0.3 eV, but it disappears in 1FUVS310.

The nature of bonding and its quantitative assessment is important in the electronic structure study of the material. While most of the MD studies interpret the bonding analysis based exclusively on the geometric distance configuration, OLCAO can quantify the strength of each bonded pair by providing the BO value. We calculate the TBO values in all 4 models and resolve them into partial pair-resolved BO (PBO) components. This is an unorthodox approach that generalizes the bonding between individual bonded pairs simply and effectively to reveal the hidden details which cannot be extricated by other methods, either experimental or computational.

In general, the BO value scales inversely with BL; however, the actual BO value of a bonded pair depends on its bonding environment and the atoms surrounding it. The calculated values of TBO and PBO are listed in Table 1, showing that at 310 K the TBO value decreases in the dry model, resulting from the elongation of some bonds. The TBO value of the solvated model increases at 310 K, opposite to the case of the dry model, which is due to the formation of more bonds between the peptide backbone and water at this elevated temperature. Figure 4 shows the comparison of PBO values for each bonded pair in simulated models at 0 K and 310 K in the form of a vertical bar diagram. The scale breaks are used along the vertical axis to delineate the minor differences between PBO values and show their magnified image. Figure 4 shows that there is a minor change in the PBO values due to the rise of temperature, except in the case of the O–H pair in the solvated model. The PBO for the O–H pair increases sharply in the solvated model at 310 K as compared with the model at 0 K, which arises due to the interaction of more water molecules with the peptide, and an increase in the HBs population at 310 K. It should be emphasized again that the small changes in these numbers at different temperature can be significant in the context of the human body.

We provide further information on the strength of individual bonded pairs presenting a complex distribution of BO versus BL in Figure 5. It clearly shows a wide variety of bonding pairs ranging from a strong covalent bond originating from N–H, N–C, C–H, C–C, C–O, and O–H pairs to weaker covalent bonds at longer BL. HBs are extremely weak but not entirely negligible because of their sheer numbers. The first group of strong covalent bonds arises from N–H, C–H, and O–H pairs with BL less than 1.2 Å. The next group of covalent BO pairs arises from C–C, C–O, and N–C bonds, whose BL lies in the range 1.2–1.7 Å. The C–C bonds show two clusterings, one having a high BO greater than 0.5 e from C–C double bonds, whereas the other group has a BO value less than 0.4 e from single C–C bonds. The BO values for N–C pairs also exhibit a group separation, having BO values greater than 0.4 e and less than 0.4 e.

The HBs are distributed in the range of 1.5–2.5 Å, with some O–H pairs having relatively higher BO values up to 0.1 e. In solvated models, the covalent O–H bonds within the water molecule have BO values higher than 0.2 e, while those bonds originating from the next neighbor or weak HBs beyond 2.5 Å have almost zero BO values. The bonding distributions due to C–S and S–S pairs have BO values comparable or slightly higher than those of strong HBs. The S–S bonding configurations are important in peptide structure, and they show significant changes in BO values due to the rise of temperature and solvation. Especially in the solvated model at 310 K, the BO of S–S bonds has a large difference. Wang et al. studied the influence of disulfide bond on properties of RGD peptide using MD and found that this bond causes the restriction on peptide mobility and affects its dipole moment [64]. The bonded pairs H–H and H–S with BL larger than 2.5 Å have BO values close to zero.

### 3.3. Hydrogen Bonding Analysis

Hydrogen bonding is especially important in a biomolecular system since it provides the key information to understand many intriguing phenomena, making the detailed analysis of HBs in the solvated models crucial. Unfortunately, in most MD studies, the analysis of HBs is carried out based exclusively on structural data, such as the BL and the location of HBs, failing to provide a quantitative assessment of HB strength and their local bonding arrangements. On the contrary, we bring forth quantitative information and deeper analysis of HBs using ab initio calculation in terms of BO-BL plot, as shown in Figure 5. It shows there exists some strong HB with BO value ~ 0.1 e, while most of the HBs have BO less than 0.1 e. At the body temperature, the overall strength of HB from the O–H bonding pair slightly decreases in the dry model while it increases in the solvated model.

In the case of solvated models, the change in the HB pattern is more complicated due to the presence of H_2_O molecules which increase the population and strength of O–H bonds at 310 K. The more HBs imply the stronger binding force of biomolecule with solvent. There is one HB originating from the N–H pair within the peptide backbone whose BO value at 310 K increases in the dry model and decreases in the solvated model because of the rotation and changes in the configuration of the peptide structure. The results on HB analysis in solvated models indicate a significant difference in the human body as compared with the vicinal environment of a dry peptide at zero temperature.

### 3.4. Partial Charge Distribution

The PC distribution of biomolecules is important for determining the long-range electrostatic and polar interaction involving the molecule. It entails a deep qualitative understanding of the structure and reactivity of a molecule. Rather than just making a rudimentary judgment by dividing PC into positive, negative, and neutral as adopted in many simplified calculations, we provide a quantitative assessment of PC of every atom in the simulated models. As the PC gives the quantity of charge transfer in atomic interaction, a positive value implies a loss, and a negative value means the gain of electronic charge.

The distribution of calculated PC for each atom in dry and solvated models is shown in Figure 6 and Figure 7, respectively, which provide a wealth of information that corroborates the electronic structure results discussed above. The solid symbols represent PC of ions at 0 K, while the hollow symbols represent PC at the body temperature. O (P) and H (P) denote O and H atoms from the peptide backbone while O (W) and H (W) denote those from the water molecule. In both dry and solvated models at 310 K, small changes appear in the PC of ions compared with their values in corresponding models at 0 K. Figure 6 and Figure 7 show that H and S lose charges, N gains charge, while C is both losing or gaining charges. In the dry model, O gains charge, while in the solvated model (Figure 7), few O atoms from the peptide backbone lose charge, and the remaining gain charge. The calculated PC values for each atom are scattered in a certain range rather than clustering and overlapping to each other, which implies the PC value is not the same even for the same atom in the same model, as their values depend on the local bonding environment. The PC of O and H in water and peptide backbone is not the same, implying that their local bonding characteristics are completely different (Figure 7). The dispersed distribution of calculated PC for each atom implies that the quantum mechanical calculation is essential to deal with such type of problem rather than using MD studies in which the PC values are predefined and kept constant in the simulation where the local environment changes.

### 3.5. Dielectric Response

The dielectric response of biomolecules in an aqueous solution has been studied for decades [65,66,67] and remains a subject of active research. The static dielectric constant of peptide/protein plays a crucial role in the calculation of the electrostatic field obtained by solving the Poisson–Boltzmann equation [68]. However, it is interesting to note that the dielectric properties of the 1FUV are seldom reported. We present a detailed analysis of the imaginary part, ε_2_ (ℏω), of the frequency-dependent dielectric function, based on the ab initio calculation under random phase approximation such as those used in inorganic crystals and glasses [69,70,71,72].

Figure 8a shows the spectrum for ε_2_ (ℏω) in 1FUV0 calculated using OLCAO, displaying a sharp peak at ~5.70 eV and one broad peak at ~14.30 eV. A similar spectrum is observed in the dry model at 310 K; however, the broad peak position shifts slightly to the lower energy side. In solvated models, in addition to the two peaks observed in dry models, there appears another sharp peak at ~0.1 eV, but with slightly less intensity (Figure 8b). One of the striking features observed is that at the body temperature, the absorption end of ε_2_ (ℏω) shifts to lower energy which is consistent with the decrease in the HOMO-LUMO gap at 310 K. We want to make it clear that our aim in this work is to compare the full dielectric spectra at 0 K and body temperature in case of a dry and solvated environment and the actual calculation of the dielectric constant is out of the scope of this work.

## 4. Discussions and Summary

In this section, we discuss the results presented above and summarize the main findings to answer the queries raised in the introduction. More accurate AIMD and DFT calculations adopted in this work provide us a wealth of quantitative information on the fundamental properties of 1FUV, which are not yet available or cannot be attained by an experimental protocol. Our findings provide a comparative analysis of the structure and properties of 1FUV and a better understanding of its solvation properties at 0 K and body temperature, 310 K. The key message is that there are discernible changes in the peptide structure and properties at 310 K, which could point to the potential implications for human health issues if such calculations can eventually be applied to large biomolecular systems.

Our results show noticeable changes in the conformation of amino acids at body temperature than at 0 K. The major structural differences observed in dry and solvated peptides at body temperature help in designing the specificity of integrins. The disulfide bridge connectivity greatly determines the orientation of the RGD motif and peptide structure which may dictate selectivity towards different integrins. The HOMO-LUMO gap of 1FUV decreases due to the rise of temperature and solvation effect. Such information is quite helpful to understand the chemical reactions involving biomolecules.

The bonding analysis is a significant result that helps to understand the interatomic interaction within 1FUV and other biomolecules. The TBO value decreases in the dry model while it increases in the solvated model at 310 K. The dry model at 0 K is more cohesive than that at 310 K, characterized by a higher TBO value, while it is just the opposite in the solvated models. The BO-BL plot analysis provides a detailed picture of interatomic bonding within the peptide structure and its explicit solvent environment. The solvated model shows a considerable peptide-water interactions occur at 310 K, increasing the HBs population than at 0 K. The detailed analysis of HBs and quantification of their strength is crucial in understanding the biological interactions inside the human body and for the design and delivery of RGD peptide-targeted drugs. The partial charge analysis provides another set of significant parameters to define the reactivity of biomolecules and their interactions with the integrins. It shows that the PC value strongly depends on the temperature as well as the solvated environment. Furthermore, the calculated PC values scatter in a certain range, even for the same atom in the same model. Quantum mechanical calculation is, therefore, absolutely necessary to capture all these crucial differences and to provide more accurate results inaccessible to classical MD studies.

Another important result is the calculation of the dielectric response function and its relation to the dielectric constant. Although this topic has attracted researchers from various backgrounds for a long time, unified and reliable ab initio calculations are still out of reach. As an initial assessment, we analyze the imaginary part of the dielectric function of the 1FUV peptide and study the differences in the dielectric spectra due to the rise in temperature and the solvation effect. Such optical spectra are helpful to estimate the long-range van der Waals–London interaction in biomolecules, as shown in the study of carbon nanotubes [73]. Overall, our results show apparent differences in bonding, PC, and dielectric function between the 1FUV at 0 K and the body temperature in a dry and solvated environment. These results can be quite helpful to develop and refine the force field parameters used in MD simulation. Unfortunately, as there is a lack of theoretical studies in 1FUV and the focus of the existing experimental study [16] is different, we are unable to compare our results with the previous results. However, we anticipate that our results will inspire future experimental and/or theoretical works to investigate further the 1FUV and related peptides, including the effect of salts and different numbers of water molecules, which significantly influence peptide structure and properties.

In the past, we have successfully demonstrated the applicability of AIMD in the study of oxide glassy materials and fluoride salts [74,75,76,77]. Our accomplishment in the present work is applying AIMD to a relatively small peptide structure that looks quite promising. It ensures the feasibility and encourages a further study of other possibly more complex biomolecules by using AIMD, of course assuming sufficient computing resources to be available.

## Figures and Tables

**Figure 1 polymers-13-03434-f001:**
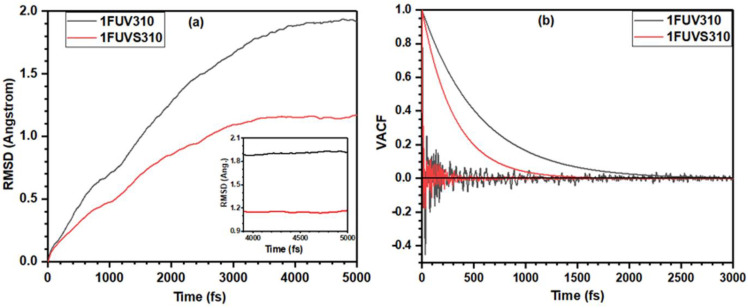
Calculated (**a**) RMSD and (**b**) VACF including exponential fitting, of simulated dry and solvated 1FUV peptide at 310 K.

**Figure 2 polymers-13-03434-f002:**
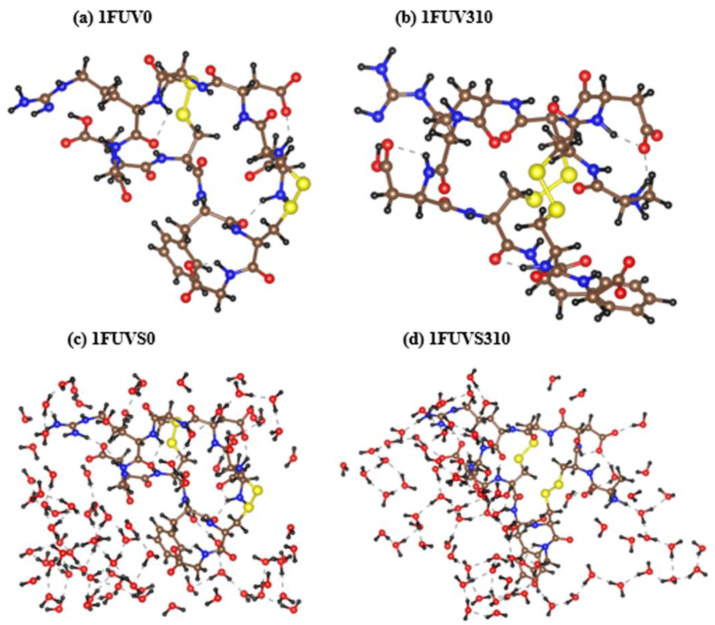
Structures of dry (**a**,**b**) and solvated (**c**,**d**) 1FUV peptide at 0 K and 310 K, respectively. (N = Blue, C = Brown, O = Red, H = Black, and S = Yellow).

**Figure 3 polymers-13-03434-f003:**
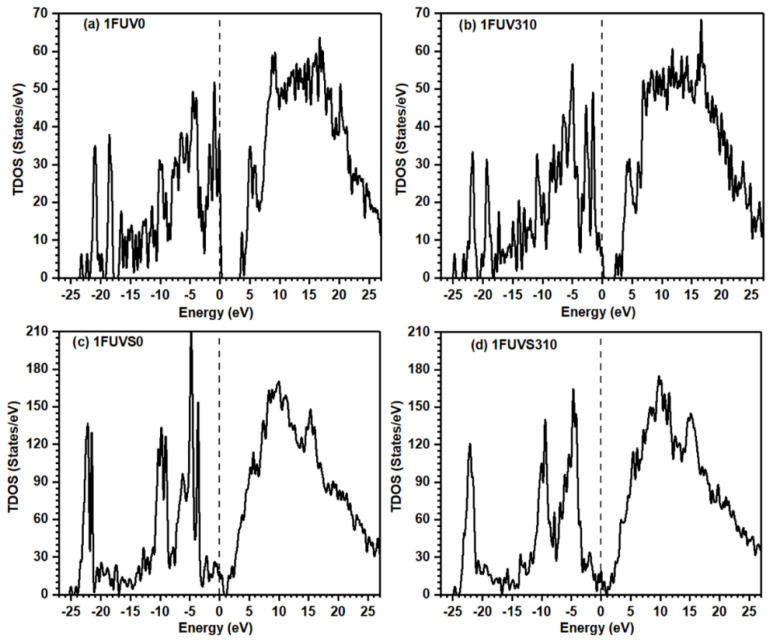
Calculated TDOS for dry and solvated 1FUV peptide at 0 K and 310 K.

**Figure 4 polymers-13-03434-f004:**
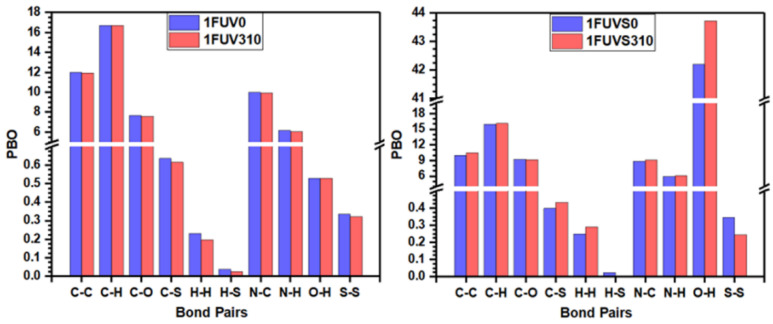
Comparison of PBO values for each bonded pair in dry and solvated 1FUV at 0 K and 310 K.

**Figure 5 polymers-13-03434-f005:**
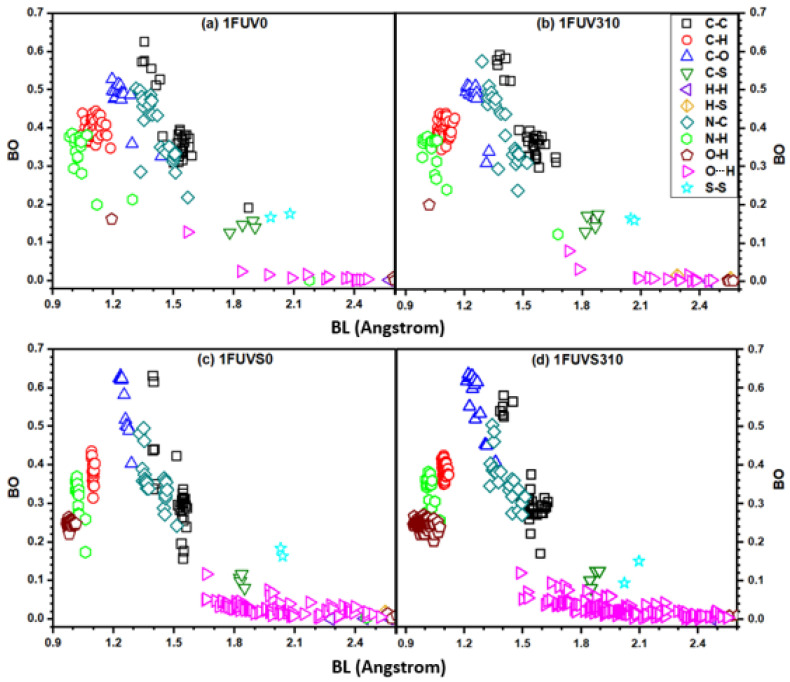
BO versus BL plot for dry and solvated 1FUV at 0 K and 310 K.

**Figure 6 polymers-13-03434-f006:**
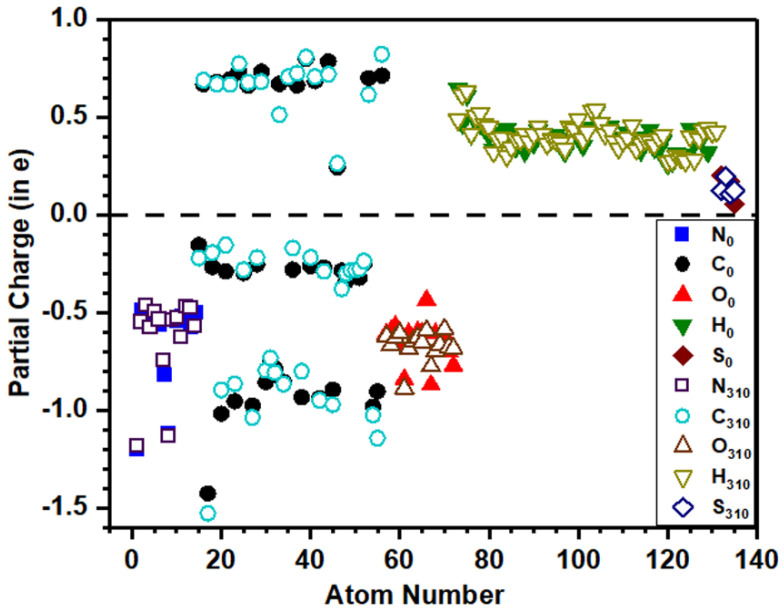
Calculated partial charge for each atom in dry 1FUV at 0 K and 310 K. (The subscripts 0 and 310 denote the temperature 0 K and 310 K, respectively).

**Figure 7 polymers-13-03434-f007:**
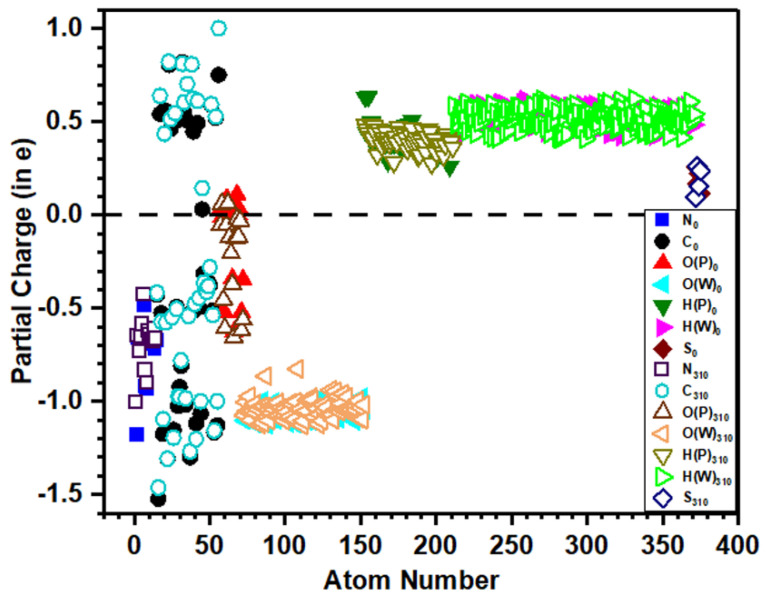
Calculated partial charge for each atom in solvated models of 1FUV at 0 K and 310 K. (The subscripts 0 and 310 denote the temperature 0 K and 310 K, respectively).

**Figure 8 polymers-13-03434-f008:**
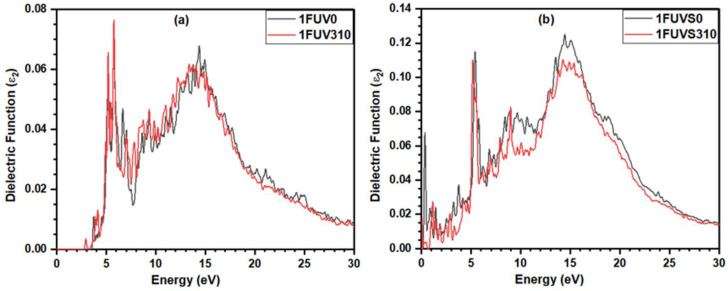
Calculated imaginary part of the complex dielectric function for (**a**) dry and (**b**) solvated 1FUV at 0 K and 310 K.

**Table 1 polymers-13-03434-t001:** Calculated TBO and PBO values for each bonded pair in simulated models.

Models	TBO	PBO										
		C‒C	C‒H	C‒O	C‒S	H‒H	H‒S	H‒O	N‒C	N‒H	N‒O	S‒S
1FUV0	54.272	11.991	16.680	7.669	0.637	0.232	0.037	0.529	9.979	6.181	0.001	0.336
1FUV310	53.887	11.944	16.705	7.581	0.617	0.198	0.027	0.530	9.926	6.034	0.000	0.323
1FUVS0	93.166	9.990	15.947	9.234	0.400	0.251	0.025	42.207	8.859	5.905	0.003	0.346
1FUVS310	95.663	10.479	16.119	9.125	0.434	0.292	0.005	43.724	9.111	6.119	0.000	0.246

## Data Availability

Data that supports the results in this study are available from the corresponding author upon reasonable request.

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
