# Peer review of "Solvent Effect on the Structure and Properties of RGD Peptide (1FUV) at Body Temperature (310 K) Using Ab Initio Molecular Dynamics"

_polymers, 2021, doi:10.3390/polym13193434_

Round 1
Reviewer 1 Report
This manuscript, written by Baral et al., uses ab initio molecular dynamics simulation to study the solvent effect on the properties of RGD peptide, such as partial charges, at the human body temperature. The paper is well-written and the introduction provides sufficient background on the topic. This work is interesting and clearly presented, and should be published.
I have only a few comments and questions, not necessary in order to have this work published here, which might improve the manuscript if addressed.
- The relaxation time of the structure can be extracted from Figure 1(b) by fitting an exponential function. This way the difference between the two curves can be compared more quantitatively.
- Figure 2 could be more useful if you would show the two structures of the peptide aligned on top of each other so that the difference between them becomes more visible.
- Is the low temperature arbitrarily chosen for the sake of comparison or is there a specific reason you decided to do simulations at this temperature? For example, why not consider the room temperature as the low temperature?
- It would be easier to compare the solvent effect on the partial charge if the two panels of Figure 7 were combined in one panel similar to the authors’ previous study published in Scientific Reports in 2014.
- Will results be affected by the presence of salt ions in the solvent?
- There are some relevant references you could include in your introduction such as:
- Molecular Simulation of Conformational Pre-Organization in Cyclic RGD Peptides by Wakefield et al. 2015
- Structural Insights into How the MIDAS Ion Stabilizes Integrin Binding to an RGD Peptide under Force by Craig et al. 2004.
Author Response
Comments:
This manuscript, written by Baral et al., uses ab initio molecular dynamics simulation to study the solvent effect on the properties of RGD peptide, such as partial charges, at the human body temperature. The paper is well-written, and the introduction provides sufficient background on the topic. This work is interesting and clearly presented and should be published.
I have only a few comments and questions, not necessary in order to have this work published here, which might improve the manuscript if addressed.
Author response:
We appreciate Reviewer 1’s positive assessment of our manuscript. Further, we thank the reviewer for providing constructive comments to improve our manuscript, which we have addressed below point by point and revised the manuscript accordingly.
1) The relaxation time of the structure can be extracted from Figure 1(b) by fitting an exponential function. This way the difference between the two curves can be compared more quantitatively.
Author response:
As suggested by the reviewer, we have fitted an exponential function in Figure 1(b) and made the following modification in the revised manuscript (page 4):
Figure 1 (b) shows the velocity autocorrelation function (VACF), including exponential fitting, during the equilibration of dry and solvated structures at 310 K, which confirms the loss of initial velocity from the previous configuration and attest to sufficient equilibration of the systems. It also implies that the amplitude of VACF is smaller in the solvated model and dies out quicker than that in the dry model. In addition, the exponential fitting shows the solvated model reaches equilibrium faster than the dry model.
2) Figure 2 could be more useful if you would show the two structures of the peptide aligned on top of each other so that the difference between them becomes more visible.
Author response:
As the amino acids are not only rotated but also their positions are changed, which makes the figure with alignment on the top more complex and busier. It will be even more complicated in solvated models where more water molecules interact with the peptide at 310 K. Therefore, we respectfully disagree with the reviewer’s suggestion and believe that the presented figure is simpler to understand the simulated peptide structures.
3) Is the low temperature arbitrarily chosen for the sake of comparison or is there a specific reason you decided to do simulations at this temperature? For example, why not consider the room temperature as the low temperature?
Author response:
In this work, our focus is to study the structure and properties of dry and solvated 1FUV at body temperature (310 K) and compare the results with a lower temperature model, 0 K in this study because of static DFT calculations. We can choose a low temperature at any value smaller than 310 K since increasing or decreasing even a unit degree temperature significantly affects human health issues. Such studies need molecular dynamics calculations at each temperature stage, and the purpose will be different than this study. We re-write our statements in the 1st paragraph of the introduction to emphasize that our focus is on the body temperature, and which reads as follows (pages 1 & 2):
The increased application of integrins in drug development and their functions in the physiological processes requires a complete understanding of the structure and properties of the RGD peptides at body temperature (310K), crucial in the design of selective inhibitors. Therefore, a detailed study of the structure and properties of 1FUV in RGD peptides at 310 K is of particular interest in delineating the modifications that occur with a rise or fall of body temperature.
4) It would be easier to compare the solvent effect on the partial charge if the two panels of Figure 7 were combined in one panel similar to the authors’ previous study published in Scientific Reports in 2014.
Author response:
As suggested by the reviewer, we have presented Figure 7 (also Figure 6) in a new way and modified the manuscript accordingly. The revised manuscript reads as follows (pages 9 & 10):
The distribution of calculated PC for each atom in dry and solvated models is shown in Figures 6 and 7, respectively, which provide a wealth of information that corroborates the electronic structure results discussed above. The solid symbols represent PC of ions at 0 K, while the hollow symbols represent PC at the body temperature. O(P) and H(P) denote O and H atoms from the peptide backbone, while O(W) and H(W) denote those from the water molecule. In both dry and solvated models at 310 K, small changes appear in the PC of ions compared to their values in corresponding models at 0 K.
5) Will results be affected by the presence of salt ions in the solvent?
Author response:
The presence of salts obviously affects the peptide structure and properties, which is out of the focus of this study. However, to introduce the importance of such studies, we add the following sentences at the end of the 4th paragraph of the discussion section (page 11).
However, we anticipate that our results will inspire future experimental and/or theoretical works to investigate further the 1FUV and related peptides, including the effect of salts and different numbers of water molecules, which significantly influence peptide structure and properties.
6) There are some relevant references you could include in your introduction such as:
- Molecular Simulation of Conformational Pre-Organization in Cyclic RGD Peptides by Wakefield et al. 2015
- Structural Insights into How the MIDAS Ion Stabilizes Integrin Binding to an RGD Peptide under Force by Craig et al. 2004.
Author response:
We have added the suggested references in the revised manuscript, and they are Refs. 19 and 36.
- Craig et al. Structural insights into how the MIDAS ion stabilizes integrin binding to an
RGD peptide under force. Structure 2004, 12, 2049-2058.
- Wakefield et al. Molecular simulation of conformational pre-organization in cyclic RGD
peptides. Journal of chemical information and modeling 2015, 55, 806-813.
Reviewer 2 Report
The work entitled “Solvent effect on the structure and properties of RGD peptide (1FUV) at body temperature (310 K) using ab initio molecular dynamics” by Baral et al. analyses the effect of temperature and solvent in the structure of RGD sequences on the 1FUV peptide using accurate ab initio molecular dynamics and density functional theory calculations. The work is well introduced, and the methodology is very detailed and complete. The readers will have no issue following the train of thought of the authors. More importantly, their results are very clearly put together, with a sequence of information that is easily followed. However, imaging requires improvement. Some schemes and graphics quality is very low and sometime makes it difficult to comprehend the information. The biggest weakness of the work is in the discussion portion. Even though the authors related well the collected information they barely criticized it or even tried and support some of the conclusions drawn from the work. It would be most important if that was made, so the novelty and the hypothesis raised and the achieved data’s quality could be improved. There are some English writing details that also require the authors attention along the manuscript; however they in no way limit the comprehension of the work done. Overall, the work is scientifically sound and should be considered for publication after major revisions are implemented in the discussion portion of the work.
Author Response
Comments:
The work entitled “Solvent effect on the structure and properties of RGD peptide (1FUV) at body temperature (310 K) using ab initio molecular dynamics” by Baral et al. analyses the effect of temperature and solvent in the structure of RGD sequences on the 1FUV peptide using accurate ab initio molecular dynamics and density functional theory calculations. The work is well introduced, and the methodology is very detailed and complete. The readers will have no issue following the train of thought of the authors. More importantly, their results are very clearly put together, with a sequence of information that is easily followed. However, imaging requires improvement. Some schemes and graphics quality is very low and sometime makes it difficult to comprehend the information. The biggest weakness of the work is in the discussion portion. Even though the authors related well the collected information they barely criticized it or even tried and support some of the conclusions drawn from the work. It would be most important if that was made, so the novelty and the hypothesis raised and the achieved data’s quality could be improved. There are some English writing details that also require the authors attention along the manuscript; however they in no way limit the comprehension of the work done. Overall, the work is scientifically sound and should be considered for publication after major revisions are implemented in the discussion portion of the work.
Author response:
We appreciate Reviewer 2’s positive assessment of our manuscript and thank the reviewer for constructive comments to improve our manuscript. Most of our figures are a combination of 2 or more individual figures, and we have tried to present them with sufficiently high resolution and in a clear way. Although individual figures are of high quality, the graphics quality drops slightly in the combined figure. In the revised manuscript, we have modified Figures 6 and 7. If there is a particular figure that needs to be changed/improved, we will feel happy to improve it if the reviewer points out such figure.
We have carefully checked the English writing/grammar and improved it in the revised manuscript. In addition, following the reviewer’s suggestion, we have enhanced our discussion section, which reads as follows [2nd, and 3rd paragraphs in the old manuscript à 2nd - 4th paragraphs in the revised one, Page 11]:
Our results show noticeable changes in the conformation of amino acids at body temperature than at 0 K. The major structural differences observed in dry and solvated peptides at body temperature help in designing the specificity of integrins. The disulfide bridge connectivity greatly determines the orientation of the RGD motif and peptide structure which may dictate selectivity towards different integrins. The HOMO–LUMO gap of 1FUV decreases due to the rise of temperature and solvation effect. Such information is quite helpful to understand the chemical reactions involving biomolecules.
The bonding analysis is a significant result that helps to understand the interatomic interaction within 1FUV and other biomolecules. The TBO value decreases in the dry model while it increases in the solvated model at 310 K. The dry model at 0 K is more cohesive than that at 310 K, characterized by a higher TBO value, while it is just the opposite in the solvated models. The BO–BL plot analysis provides a detailed picture of interatomic bonding within the peptide structure and its explicit solvent environment. The solvated model shows a considerable peptide–water interactions occur at 310 K, increasing the HBs population than at 0 K. The detailed analysis of HBs and quantification of their strength is crucial in understanding the biological interactions inside the human body and for the design and delivery of RGD peptide-targeted drugs. The partial charge analysis provides another set of significant parameters to define the reactivity of biomolecules and their interactions with the integrins. It shows that the PC value strongly depends on the temperature as well as the solvated environment. Furthermore, the calculated PC values scatter in a certain range, even for the same atom in the same model. Quantum mechanical calculation is, therefore, absolutely necessary to capture all these crucial differences and provide more accurate results inaccessible to classical MD studies.
Another quite important result is the calculation of the dielectric response function and its relation to the dielectric constant. Although this topic has attracted researchers from various backgrounds for a long time, unified and reliable ab initio calculations are still out of reach. As an initial assessment, we analyze the imaginary part of the dielectric function of 1FUV and study the differences in the dielectric spectra due to the rise in temperature and the solvation effect. Such optical spectra are helpful to estimate the long-range van der Waals-London interaction in biomolecules, as shown in the study of carbon nanotubes [73]. Overall, our results show apparent differences in bonding, PC, and dielectric function between the 1FUV at 0 K and the body temperature in a dry and solvated environment. These results can be quite helpful to develop and refine the force field parameters used in MD simulation. Unfortunately, as there is a lack of theoretical studies in 1FUV and the focus of the existing experimental study [16] is different, we are unable to compare our results with the previous results. However, we anticipate that our results will inspire future experimental and/or theoretical works to investigate further the 1FUV and related peptides, including the effect of salts and different numbers of water molecules, which significantly influence peptide structure and properties.
- Rajter et al. Calculating van der Waals-London dispersion spectra and Hamaker coefficients
of carbon nanotubes in water from ab initio optical properties. Journal of Applied Physics
2007, 101, 054303.
- Assa-Munt et al. Solution structures and integrin-binding activities of an RGD peptide with
two isomers. Biochemistry 2001, 40, 2373-2378.
Round 2
Reviewer 2 Report
The authors have implemented significant alterations in the manuscript and it is now ready for publication.